# EMBER—Embedding Multiple Molecular Fingerprints for Virtual Screening

**DOI:** 10.3390/ijms23042156

**Published:** 2022-02-15

**Authors:** Isabella Mendolia, Salvatore Contino, Giada De Simone, Ugo Perricone, Roberto Pirrone

**Affiliations:** 1Dipartimento di Ingegneria, Università degli Studi di Palermo, 90133 Palermo, Italy; isabella.mendolia@unipa.it (I.M.); roberto.pirrone@unipa.it (R.P.); 2Molecular Informatics Group, Fondazione Ri.MED, 90133 Palermo, Italy; gdesimone@fondazionerimed.com

**Keywords:** deep learning, drug design, virtual screening, embedding

## Abstract

In recent years, the debate in the field of applications of Deep Learning to Virtual Screening has focused on the use of neural embeddings with respect to classical descriptors in order to encode both structural and physical properties of ligands and/or targets. The attention on embeddings with the increasing use of Graph Neural Networks aimed at overcoming molecular fingerprints that are short range embeddings for atomic neighborhoods. Here, we present EMBER, a novel molecular embedding made by seven molecular fingerprints arranged as different “spectra” to describe the same molecule, and we prove its effectiveness by using deep convolutional architecture that assesses ligands’ bioactivity on a data set containing twenty protein kinases with similar binding sites to CDK1. The data set itself is presented, and the architecture is explained in detail along with its training procedure. We report experimental results and an explainability analysis to assess the contribution of each fingerprint to different targets.

## 1. Introduction

Drug discovery is a very long and expensive process that includes many stages such as drug target identification, target validation, virtual screening (VS), hit-to-lead generation, lead optimization, and so on [1]. Moreover, developing a new drug has a mean pretax expenditure above 2 billion USD and takes about 10–15 years [2,3]. Despite the huge investment of time and money, the estimated clinical approval success rate of innovative small molecules during the drug discovery process is about 13%; thus, the overall risk of failure is very high. Drug design is supported by computational methods in almost every stage. Yu and MacKerell [4] report a review that describes the drug discovery process and the corresponding computer-aided drug design methods. Computational methods do not guarantee a systematical assessment of molecular characteristics (e.g., bioactivity, *ADMET* properties, selectivity, and physicochemical properties) but generate lead molecules with favourable properties in silico.

In particular, Virtual Screening (VS) is an often discussed topic in Chemoinformatics and Medicinal Chemistry and is widely applied in pharmaceutical research. VS consists of screening large small-molecule databases searching for bioactive molecules with respect to the target under investigation. This enables the researcher to cut the cost of experimentally testing thousands of compounds through a severe reduction in the number of candidate molecules. Research in the field of VS gained increasing importance in the last decade when Deep Learning (DL) became a mature discipline [5]. In this field, the scientific debate is very rich with respect to the proper method for representing molecular structures that are learned by the network. The very first architectures used classical representations such as molecular fingerprints [6] and SMILES notation [7]. Recently, molecular graphs have been investigated along with neural embeddings, essentially a learned low-dimension vector representation for discrete and/or categorical data. Embeddings can be used suitably to train a neural network in place of the original samples. In fact, neural embeddings represent a way to fit the input data representation in the numerical constraints posed by the training procedures of a neural model. On the other hand, embeddings mask the original features of the input data, and it is hard to devise an explanation of the model behaviour. Explainable AI (XAI) is aimed at providing a description of how the model uses features to build its predictions, and this is a crucial topic to make extensive uses of neural models viable in the general context of life sciences. In this work, we propose EMBER, a simple but novel neural embedding for molecular structures that allows explainability in a multitarget VS task. The major contributions of the presented study are reported in the following:The EMBER (EMBedding multiplE molecular fingeRprints) embedding is proposed, which is made by multiple molecular fingerprints that have been generated using complementary methods to search for molecular substructures and are stacked as the spectra of a sort of “molecular image”; such an embedding aims at exploiting the ability of Convolutional Neural Networks (CNN) in learning the proper features, as they do for images;A multi classifier has been developed to prove the previous claim, which performs very well in screening ligands on twenty protein kinases presenting the closest binding sites to CDK1; moreover, our architectural design lowers the parameter numberlA curated data set made by nearly 90,000 ligands labeled as active/inactive against 20 Kinase target selected as the most similar to CDK1.An explainability analysis has been performed to assess the most relevant features for the classification task, and the results of this analysis confirm some very recent in vitro studies that outline the relevance of pharmacophore-like description fingerprints when addressing bioactivity classification for kinase inhibitors

Our study is part of a wider research study aimed at screening novel compounds with anti-cancer properties.

In cancer therapeutics, it is well known that protein kinases are key regulators of cell function and constitute one of the largest and most functionally diverse protein families. By adding phosphate groups to the substrate, they regulate the activity, localization, and overall function of many proteins and serve in orchestrating the activity of almost all cellular processes. Kinases are involved in signal transduction and coordination of complex functions, such as the cell cycle. Kinases have been considered one of the most promising target family in oncology. In the search of the best protein family to be focused on, the choice is usually guided by the possibility to have mutated targets that can be used for selectivity issues. Mutated kinases can become active by constitution and, thus, cause diverse cellular anomalies resulting in cancer initiation or growth. One of the most well-known mutated kinase example is BRAF, which is frequently mutated on Val-600 (p.V600E) and is a driver mutation in several cancers, including colorectal cancer, melanoma, and thyroid cancer [8].

Between different kinases, CDK1 is a central regulator that drives cells through G2 phase and mitosis. Diril et al. [9] generated a conditional-knockout mouse model to study CDK1 functions in vivo. From this study, it was found that the low presence of CDK1 in the liver confers complete resistance against tumorigenesis induced by activated RAS and P53 silencing. Considering the large role of cycline-dependant kinases in the tumor progression, our research group has been focusing all efforts on the design of new CDKs inhibitors, with a special focus on CDK1.

### 1.1. Theoretical Remarks

In this section, some remarks are reported about both Deep Neural Networks for Virtual Screening and molecular embeddings.

#### 1.1.1. Deep Neural Networks for Virtual Screening

In the last few years, DL has been used in all research fields related to life sciences: Angermueller et al. [10] report a review of DL techniques in computational biology, while Anwar et al. [11] propose a comprehensive report on DL for medical imaging. DL solutions have been proposed in support of all stages in drug design workflows [12]; moreover, in general, AI-based techniques such as Decision Support Systems and robotic platforms are expected to be in synergy with the human medical chemist, in the near future, in order to perform drug discovery [13]. Virtual Screening is undoubtedly one of the most investigated topics for DL applications. The reader is referred, in particular, to the work by Kimber et al. [14] for structure-based approaches and to the paper by Sydow et al. [15] for ligand-based ones. The very first DNN for QSAR prediction was a multi task classifier presented by Dahl et al. [16] where the same candidate was tested for its bioactivity on different assays. Wallach et al. [17] presented *AtomNet*, which is regarded as the first CNN for structure-based screening. Duvenaud et al. [18] proposed a CNN to learn circular fingerprints from molecular graphs, and some experiments were carried out to prove their efficacy both in solubility and drug efficacy prediction. Pereira et al. [19] proposed *DeepVS*: Such a CNN makes use of the notion of the *context* of an atom in the protein–compound complex that is a vector representation of the structural properties of its neighborhood. Hiroara et al. [20] used SMILES notation by describing the compound to create a *feature matrix* where each column is a one-hot encoding of the presence of a particular SMILES symbol at a certain position. Such a representation is fed to a CNN in order to detect “chemical motifs” that are relevant binding substructures.

#### 1.1.2. Molecular Embeddings

The presented review outlines that a great effort in DL applications for VS has been devoted to arranging all molecular representations as CNN input tensors. In recent years, the debate on DL applications relative to drug discovery has broadened: The topics related to model explainability attracted the attention of the scientific community [21]. In the context of eXplainable AI (XAI), using the proper input representation is a crucial topic for assessing model behavior. In fact, a direct description of the molecular structure in the form of graphs or SMILES allows the immediate comprehension of the chemical meaning of model results. In this respect, many recent approaches to drug design tasks are oriented to nonconvolutional models. De novo drug design exploits the use of Long Short-Term Memory cells (LSTM) [22] or transformers [23] to process molecular information. Bjerrum [24] proposed LSTMs for QSAR learning from SMILES. All these approaches allow for model explainability by *feature attribution*, which is building a function of the model inputs r=E(x), where $r_{i}$ is a relevance score of the *i*-th feature $x_{i}$.

Moreover, molecular graphs are very explainable input descriptions because they represent molecules in an immediate manner, but GNN architectures, in general, make an extensive use of the notion of *embedding*. Neural network embeddings were introduced first in the context of NLP [25] and are one of the most remarkable findings in DNN research. A neural embedding is a continuous vector representation for discrete and/or categorical variables and is used with the two-fold purpose of reducing the dimensionality of the original input space and achieving better numerical stability in activation functions that are designed to cope with small real values. A neural embedding can be computed explicitly by using a function that maps the inputs to a vector representation, but they are learned through a proper DNN in many cases. A molecular graph conveys structural information through both categorical and numeric data at each node or edge: atoms and bonds types, presence of rings, aromaticity, formal and partial charge, and so on. As a consequence, the use of embeddings is a common practice when designing a GNN for molecular analysis.

After the work by Duvenaud [18] where circular neural fingerprints were learned by molecular graphs, molecular Graph Convolution Networks were introduced by Kearnes et al. [26] where a systematic definition of suitable convolutional layers is presented to process feature vectors in either single atoms or atom pairs when a chemical bond connects them. The overall architecture uses many layers where both atom–atom and atom–pair interactions at the *k*-th layer are used to build the k+1-th layer. On top of the last atom layer, global molecular features are computed by using fuzzy membership functions representing the extent to which an atom contributes to each molecular feature. This representation, in turn, is used for classification tasks. Coley et al. [27] proposed a *representation learning* approach where a suitable “fingerprint” is learned by convolutional layers that start from single atoms’ feature vectors and use an increasing neighborhood radius to collect information from connected atoms, thus taking into account structural features depending from the chemical environment. Molecular tensors obtained from convolutions are then flattened to form a fingerprint that is used to predict aqueous solubility, octanol solubility, melting point, and toxicity. Previous GNNs belong to the family of so-called Message Passing Neural Networks (MPNN) that were introduced thoroughly by Gilmer et al. [28]. MPNNs perform learning at each state by using a “message function”, passing information between nodes along the edges, followed by an “update function” that computes the new state at each node.

Torng et al. [29] investigated drug–target interaction by using graph autoencoders. A neural embedding for target pocket features is learned by using a graph variational autoencoder (VAE) that is a DNN trained to learn a latent representation of the inputs in an unsupervised manner: two mirrored CNNs are coupled, and the overall network is trained with its inputs. The activations of the innermost layer form a low-dimensional *latent representation* of the input space. The weights of the trained encoder in the Graph-VAE are used to perform fine tuning in a target Graph-CNN that is trained in parallel to a ligand Graph-CNN. The two Graph-CNNs are fed in parallel to a fully connected “interaction” layer and then to the output binding classifier. Koge et al. [30] proposed a molecular embedding where hypergraph molecular representations are learned by VAEs based on RNNs along with a regression model for physical molecular properties so that anchor positive and negative molecular samples with respect to a particular property have a latent representation that maintains similarity. Finally, Ishiguro et al. [31] made use of the Weisfeiler–Lehman (WL) embedding of the molecular graph as the input for an MPNN. The WL embedding is a simple algorithm that enumerates the neighbors of each atom so that the input of the MPNN is formed by the atom label and the vector of its neighbors’ labels.

## 2. Results

Table 1 and Table 2 report the results of the proposed multiclassifier on the test set. In particular, Table 1 reports the accuracy and loss values obtained for single target. The overall performance of the network remains high in terms of global accuracy when analysing each single target: this finding is confirmed by high AUC values. In general, sensitivity values are low because the data set is strongly unbalanced with respect to reflecting true operational screening conditions.

Table 1 reports also the values of the Matthews Correlation Coefficient (MCC), which is a well known index used for binary classification that returns a value in [−1;1] and can be related to the chi-square statistic for a 2×2 contingency table, which is a binary classifier’s confusion matrix. In particular, the relation with chi-square statistic is expressed by $∥\mathrm{MCC}∥=\sqrt{\chi^{2}/n}$ where *n* is the number of observations so it measures the dependency of the predictions from true (i.e., expected) labels. The form of this indicator is related to the results reported in Table 2, which is devoted to explaining the actual screening capabilities of our model and contains both Enrichment Factors (EF) and True Positives versus Positives (TP/P) ratio for each target at different percentages.

EFs after x% of the focused library were calculated according to the following formula: (1)$$EF=\frac{N_{\mathrm{experimental}\;}{}^{x\%}}{N_{\mathrm{expected}\;}{}^{x\%}}=\frac{N_{\mathrm{experimental}\;}{}^{x\%}}{N_{\mathrm{active}\;}\cdot x\%}$$
where $N_{experimental}$ is the number of experimentally found active structures in the top x% of the sorted database, $N_{expected}$ is the number of expected active structures, and $N_{active}$ is total number of active structures in database [32]. EF computes the number of predicted true actives, in decreasing probability order, in a fixed percentage of the test set. Typical percentages are 5% and 10%, but in this study, we also tested the performance at 1%. Such a measure is intended to provide the number of times a particular screening procedure performs better than a pure random process.

EF values reported in Table 2 are considerably high and drop to 9 only at 10%. This result is truly remarkable even though no drug designer takes into account such large test set percentages. Moreover, all such values are considerably higher than the ones considered sufficient for a good model [32].

In order to measure explicitly the classifier’s ability to prioritize ligands, we also reported values of the ratio between the True Positives (TP), which is the number of correct predictions prioritized at the top x% of the test set, and the Positives (P), which is the total number of positives in the test set for each target. We calculated this parameter at different percentages of the test set. The use of the TP/P indicator explains some controversial EF values. The worst EF values (less than 20 at every percentage) are obtained for the JAK2 and EGFR targets, respectively. This result comes from the high abundance of active molecules in the test set that are much higher than the the number of ligands considered at each percentage. In fact, the TP/P ratio reported in the same table confirms that the classifier correctly prioritizes as many active molecules as it can for the considered test set percentage for both target proteins.

Finally, MCC values in Table 1 are in line with TP/P values, as it was expected, due to the very similar form of these indicators. In fact, the highest MCC was obtained exactly for the JAK2 target.

In order to prove the practical effectiveness of our approach, we conducted a simple experiment on ligands prioritized by our classifier for the CDK1 target. We explicitly extracted ChEMBLIDs of the top five molecules prioritized by our system in the test set and inspected both their chemical structure and their activity parameters. Table 3 reports the results, and it can be observed that all of them are strongly active against the target.

In line with our commitment to provide an explanation of the role of each fingerprint in our embedding, we conducted an analysis of our trained network using the well known SHAP framework. SHAP stands for *SHapley Additive exPlanations* [33], and it is a game-theoretic approach that was proposed first by Lipovetsky and Conklin [34]. In this work, the relevance of each predictor in a linear regression model is measured by using the *Shapley Value (SV) imputation*, which is a method that ranks the importance of each player in a multiplayer game over all possible combinations of players. The authors use *SHAP values* as the unique measure for feature relevance in an additive feature attribution explainability model, which is defined by a linear combination of the features to be explained ($z_{i}$) weighted by some importance factors ($\phi_{i}$). The SHAP value for feature $z_{i}$ is estimated as the SV ($\phi_{i}$) of a conditional expectation function $E[f\left(z\right)|z_{i}]$ describing the expected prediction over the entire feature set *z* conditioned to $z_{i}$. Both model-agnostic linear explanations and model-specific computations of SHAP values are proposed.

In our case, we adopted the so-called *Deep SHAP* explanation model that is suited for CNN because it combines SHAP values with recursive relevance score computations proposed in *DeepLIFT* [35]. The DeepLIFT explainability model assumes that a difference ($\Delta t=t-t_{0}$) in an output neuron between actual activation *t* and a reference one $t_{0}$ is related to the activation difference $\Delta x_{i}$ in whatever contributing neuron by the *summation-to-delta* property $\sum_{i}C_{\Delta x_{i}\Delta t}=\Delta t$, which is a constraint on the relevance scores ($C_{\Delta x_{i}\Delta t}$). Deep SHAP applies the DeepLIFT approach to the expectation function’s, $E[f\left(z\right)|z_{i}]$, reference value.

The results of our analysis are reported in Figure 1; on the left, we reported the SHAP values for each target and for each fingerprint averaged on the entire test set, while, on the right, the CDK1-only analysis is reported as an example of the results obtained target-by-target. Here, each fingerprint has been grouped in 64 bins to enhance readability.

As we expected, SHAP values are arranged in a manner that some fingerprints are relevant as a whole for predicting a target, while others have no contribution, which includes all SHAP values that are almost zero for each bit of the fingerprint. All targets exhibit the same relevant fingerprints even if actual SHAP values differ from each other.

## 3. Discussion

In this section, we will discuss two main topics covered in the present study that constitute the definition of EMBER and of the related data set.

EMBER is not a fixed embedding; rather, it has to be intended as a general criterion for embedding molecular fngerprints. An explainability analysis helps the researcher to tune the embedding toward the most effective fingerprints for the task at hand. In this respect, FeatMorgan and Layered and RDKIT fingerprints demonstrated a major influence on prediction results when compared to the others. We tried to rationalize this observation related to the fingerprint composition. In detail, FeatMorgan is a kind of FCFP circular fingerprint where the ligand is characterized by the functional descriptions of atoms directly related to its binding capability (e.g., hydrogen donor/acceptor, polarity, aromaticity, and so on). Most likely, for such a kind of classification that is not merely based on the chemical path but also on the ligand’s capability to bind specific protein residues, such forms of ligand description outperform when compared to the simple ECFP circular fingerprint, only relative to atom-type paths. RDKIT and Layered fingerprint are both based on substructure decomposition (e.g., aromatic rings). In a recently published work by Zhu et al. [36], the authors conducted a chemoinformatic analysis of 2139 Protein kinases inhibitors and found the majority of these molecules as “flat” with a very low fraction of ${\mathrm{sp}}^{3}$ carbons and a high number of aromatic rings. From the study, it was also demonstrated that the average weighted hydrogen bond count was inversely proportional to the number of aromatic rings. In detail, it seems that, in the binding affinity to protein kinases, there was a correlated compensation between H-bond interactions and aromatic and non-bonded interactions. Such an inverse relationship strongly suggests the importance of the balanced presence of hydrogen bond donors and acceptors and aromatic moieties within the ligand for the molecular recognition of protein kinase inhibitors.

In our opinion, the interpretation of the above-described interaction elements for kinase inhibitors is better performed by the FCFP, RDKIT, and Layered fingerprints compared to the other fingerprints mainly based on the mere description of the chemical path, and not on the pharmacophoric role of molecular elements.

With respect to our new data set we want to outline some points. We claim that the definition of a completely new publicly available data set is a valuable contribution to the scientific community for many reasons. Firstly, the data set contains curated data where both ingestion from public databases and annotation with activity data have already been performed. This data set is aimed at serving as a benchmark for novel DL approaches in VS. Moreover, it has been conceived explicitly for training DL models, while keeping in mind the size of a VS task in the real world. One of the main strengths of our approach is the use of a data set containing ligands with annotated activity data. This differentiates the approach compared to the mass use of decoys. We did not use decoys as it is well known that such data introduce bias in machine learning models toward the class they have been labeled in. Moreover, our data set contains many real samples to allow suitable training, but class balancing is purposely uneven with a 1:100 ratio between active and inactive ligands. The use of molecules annotated for more than one kinase receptor allowed our model to consider the selectivity of target assignation, as is usually performed in wet lab assays. Finally, a careful annotation procedure has been adopted where we empirically choose the thresholds for activity parameters $IC_{50},K_{I},K_{D}$ to minimize the risk that some weakly active ligands being mislabeled as inactive and vice-versa. We tried to overcome mislabeling by direct inspection of the activity values in our data using clustering to assess the effective number of classes to be used. Actually, two clusters were devised in our data with respect to the chosen activity parameter, and the relative threshold value was chosen according to the scientific literature after having verified that it is falling in between the two clusters. The only objection to our line of reasoning is that there is no guarantee that our labels are correct in every case due to an absence of certain biological information with respect to inactivity coming from assays that can exhibit reduced sensitivity.

## 4. Materials and Methods

In this section, the main idea behind EMBER is presented along with implementation details of the classifier proposed to prove its efficacy. Moreover, the data preparation procedure is reported.

### 4.1. EMBER Multi-Fingerprint Embedding

A major contribution of this work is the introduction of EMBER, an embedding that is obtained using different molecular fingerprints bundled as the “channels” of the input tensor of a 2D CNN. This section is devoted to explaining the motivations of our choice.

In our approach, molecular fingerprints are regarded as different “spectra” of the same *molecular image*. In fact, different fingerprints collect information about atomic neighborhoods using heterogeneous criteria: moving along bond-connected paths, exploring circular regions, encoding atom pairs and their bond distance, and so on. As a consequence, different fingerprints convey diverse structural information about the same molecule. Some of the authors in a recent work [37] present a comparison between different deep classifiers and ML approaches for assessing ligands’ bioactivity on Ciclyn Dependent Kinase 1 (CDK1). In that work, the authors made the same assumptions expressed here, and they used seven fingerprint families: *RDKit*, *Morgan*, *AtomPair*, *Torsion*, *Layered*, *FeatMorgan*, and *ECFP4*. The idea under fingerprint generation is to apply a kernel to a molecule for generating a bit vector. Typical kernels extract features from the molecule, hash them, and use the hash to determine bits that should be set. The typical fingerprint size range is from 1K to 4K bits: In the cited work, we used the 1024-bit size. With respect to the fingerprint types we selected, they can be grouped into two classes: *pathway-based*, also known as *topological*, and *circular*. Pathway-based fingerprints encompass RDKit, Atompair, Torsion, and Layered. In this case, the kernel is linear, and each fingerprint differs in atom types and bond types. For example, RDKit’s atom types are set by atomic number and aromaticity. In Layered, both atom and bond types contribution are determined by the particular layers included in the fingerprint. Circular fingerprints include Morgan, Featmorgan, and ECFP4. In this case, the kernel is circular and takes into account the neighborhood of each atom based on the selected radius (usually from 1 to 3).

The Morgan algorithm was presented as a method for solving the molecular isomorphism problem: identifying when two molecules, with different atom numberings, are the same. It provides numeric identifiers to each atom using an iterative process that begins with a rule that encodes the numbering invariant atom information into an initial atom identifier and ends with identifiers from the previous iteration. As a result, the created identifiers are unaffected by the atoms’ original numbering. The process is repeated until each atom’s identifier is unique.

It has often been argued that Morgan and EFCP fingerprints are the same, but it is not entirely true because the ECFP generation procedure is actually derived from the Morgan algorithm with significant improvements, especially in relation to aromatic groups. Such a process stops after a predefined number of iterations rather than when identifier uniqueness is attained. Initial atom IDs and all subsequent identifiers are grouped into a collection that determines the extended-connectivity fingerprint. The ECFP algorithm retains intermediate atom identifiers rather than discarding them. This means that the iterative process does not have to proceed all the way to the end (that is maximum disambiguation); rather, it is carried out for a fixed number of iterations. Moreover, algorithmic optimizations are available in ECFP because perfect accuracy is not necessary for disambiguation [38].

The architecture presented in the cited work exploits the contribution of each fingerprint by *parameter sharing* where seven convolutional branches are merged in a unique deep CNN, and the training procedure is in charge of merging the information conveyed by each single branch. The resulting architecture performed very well but it used 51,449,735 parameters to classify ligands on only a single target.

The lesson learned by our previous architecture is that parameter sharing provides a coupling that is too loose between inputs to achieve the high model capacity that is needed for an effective VS where multiple targets have to be taken into consideration. As a consequence, in this study, we regard the input fingerprints as the *features* of our molecular representation and use a unique deep CNN that is the ideal model for multi-channel image classification to perform our analysis. Molecular fingerprints have been widely used for many decades as a key technique in Virtual Screening, and they are no doubt an algorithmic embedding for molecular information with all the pros and cons of using such an approach. The two-phases fingerprint algorithm are as follows: At first, information is collected from an atomic neighborhood; then, hashing is used to set the actual bits in the binary string and makes a molecular fingerprint “opaque” with respect to the direct explanation of the molecular graph, even if it retains global information about the presence of particular substructures in the molecule. On the other hand, this algorithm ensures a similar computational process for each fingerprint family, and this enforces our claim about their use as channels of the same input tensor.

Finally, even if very recent research moves quickly towards learned molecular embeddings, we want to use a solid reference framework for assessing the explainabilty of our approach due to the loss of explicit structural information induced by the use of fingerprints. In this respect, our multitarget classifier will be analyzed by using a well-known framework for feature attribution, which is the standard approach in CNNs.

### 4.2. Data Preparation

The targets considered in the study were derived from the similarity approach reported below. This method consisted of the *IFPs Tanimoto Similarity* calculation for proteins with high similarity relative to CDK1 (Cyclin-dependent kinase 1). Binding site similarity was calculated on both amino acid sequences and interaction patterns with known ligands (experimental data of relative crystallography to ligand–receptor interaction). We took the top twenty proteins with a similarity coefficient, ≥0.80.

At the same time, In order to enrich inactive molecule libraries, we used the opposite of the concept of similarity, which is *dissimilarity* (similarity coefficient <0.1).

Of these twenty proteins, we have extracted a portion of data from the CheMBL molecular database [39] where the biological activity of compounds was measured mainly using the *half maximal inhibitory concentration* parameter ($IC_{50}$), which is the amount of substance needed to inhibit the target protein by one half. In order to identify the largest number of molecules, we used all other parameters available on ChEMBL, such as *inhibition constant* $K_{I}$ and *dissociation constant*
$K_{D}$. A good rule of thumb used for both $IC_{50}$ and $K_{I}$ is that values less than 1.0μM imply good bioactivity, while values greater than 10.0 μM indicate low or negligible bioactivity. The literature does not report a precise $K_{D}$ threshold to be used for labeling a compound as *active* or *inactive*. Therefore, we clustered our data by using the well known K-means algorithm with respect to the $K_{D}$ value separately for each target and devised a suitable threshold by using the well known *elbow method*. This heuristic consists in clustering data points (x) with a variable number of clusters *k* while plotting the *Within-Cluster Sum of Squares*: (2)$$WCSS=\sum_{i=1}^{k}\sum_{x\in C_{i}}{\left(x-\mu_{i}\right)}^{2}$$
where $C_{i}$ is the *i*-th cluster, and $\mu_{i}$ is its centroid. The plot will exhibit an “elbow” in correspondence of the optimal value for *k*. In this manner, we obtained k=2 for each target as it was expected, and we were also able to evaluate the centroids and the extent of each cluster. By analyzing clustering results, we obtained the value $K_{D}=7\;\mu$M as a good threshold to separate data correctly for each target.

Based on the available data in ChEMBL, the number of inactive compounds for each protein evaluated in this study was too low to build a deep learning model. The authors preferred not to use Decoys molecules for the inactives set because of some known issues about their use, especially in DL methods. Madhavi Sastry et al. [40] had already reported a variable performance of decoys based on targets and the method used for virtual screening in 2013. Then, in more recent literature, mainly focusing on the use of decoy data sets for DL has revealed some hidden biases when testing CNN virtual screening performance evaluation [41]. Moreover, Yang et al. in their recent work [42] pointed out the importance and, at the same time, the lack of publicly available DBs that are sufficiently large and unbiased data sets used for robust AI models. In addition, the work enlightened once more how the use of decoy data sets to train the model presents some critical issues. In light of these considerations and since this workflow is based on a multitarget affinity approach, the authors preferred to create their own data set starting from the ChEMBL database and exploiting dissimilarity metrics to enrich a diversity-based inactive DB. Therefore, in order to enrich the library of inactive compounds for each kinase, two different approaches were used. The first one was based on the collection of active ligands on targets presenting different ligand binding interaction patterns compared to the 20 reference ones in the study. The second one relied on the search for dissimilar compounds compared to co-crystallized kinase inhibitors.

Molecules retrieved by these two approaches were then examined to avoid the presence of duplicates. The advantage of using these two different approaches allowed the creation of a data set with a wide chemical space of compounds.

Both methods are based on a workflow built with KNIME Analytics Platform [43] (Knime version 3.7.1).

In the first approach, the idea was to identify kinases with less similar binding sites compared to the 20 targets under investigation; for each of them, active compounds were chosen. In order to perform this analysis, a workflow was built using “3D-e-Chem-KLIFS” nodes, which return information on the entire human kinome from the “Kinase-Ligand Interaction Fingerprints and Structures” database [44] (KLIFS - release version 2.4, developed by the Pharmaceutical Chemistry Division—VU University Amsterdam).

In fact, this database contains detailed information about structural kinase–ligand interactions relating to all the structures of the catalytic domains of the human protein kinases deposited in the Protein Data Bank. The *Structures Overview Retriever* node was used to obtain the structure IDs of each reference kinase and all other human kinases (total 555). All kinases data were processed as input by the *Interaction Finger print Retriever* node to generate the protein–ligand interaction (IFP) fingerprints for subsequent chemoinformatics analysis. Additionally, this node corrects fingerprints for gaps and missing debris within binding pockets, thus enabling free-for-all comparisons. Once interaction fingerprints for each protein–ligand complex were obtained, a dissimilarity analysis was performed between each kinase’s IFP by using the KNIME *Similarity Search* node. For this purpose, the Tanimoto coefficient was used as a method to calculate the distance (or dissimilarity) between each and all other human kinase IFPs. The results were also filtered, setting a coefficient range of [0–0.15]. For each kinase, a list of proteins that satisfy this dissimilarity criterion was obtained, and for each of them, the compounds considered as actives in the literature were collected. In particular, for each kinase that was dissimilar to a reference one, only compounds with *IC*_50_ values <1.0μM were collected using the ChEMBL Database v26.

Nevertheless, we wanted to further expand the number of inactive molecules by using a second approach. This second approach consists of a ligand chemical diversity search. Specifically, it was based on structurally diverse ligands compared to known active co-crystallized ligands for each protein used in this work.

The ligands in sdf format for each 3D structure of the twenty proteins (see ligand code in Table 4) were downloaded from the Protein Data Bank [45,46] (RCSB). Actives compounds were downloaded from crystal structures with a resolution less than two.

In order to enlarge our data set, 601,810 small molecules were downloaded from the entire ChEMBL DB v26 and used for dissimilarity analysis with ligands obtained from PDBs. All small molecules that were not relevant for classification purposes were removed according to the following criteria:Molecular weight > 100;Number of carbon atoms > 10;Number of nitrogen atoms > 2;Number of oxygen atoms > 2;At least one aromatic ring.

Similarity analysis was conducted by calculating the Tanimoto coefficient using ECFP4 fingerprints. Different compounds with respect to kinase inhibitors that were previously downloaded were selected from ChEMBL DB in order to possess diverse chemotypes.

As result, the use of three different methods to enrich the inactive data set allowed us to obtain a diverse set. The inactive set, in fact, was mainly composed by the inactives found on Chembl in the end to which other molecules actives on different proteins were added. Such an approach had two advantages. The first one was the possibility to have a large and diverse chemotypes space. Moreover, using these three different approaches (that is, using different approaches to select molecules), we minimized the possibility of having analogue bias and artificial enrichment typical of the usage of decoys or uncurated data sets [47].

The overall data set was built starting from two separate sets. The first one was made by 64,600 compounds that are inactive for all the targets. The second data set contains all ligands that are active on at least on one target. In the end, we merged the two data sets to obtain a final one that has a 1:100 active/inactive rate, which is referred to the less abundant class (CDK6) (see Table 4).

This final data set consisted of 89,373 molecules and was separated into a training set (68,370 molecules), test set (13,046 molecules), and validation set (7597 molecules), respectively.

The molecules were manipulated on the Knime platform in order to generate the seven fingerprints used as the channels of our embedding, which is used as input to the network.

Given the intrinsic sparsity of a molecular fingerprint, we chose to transform the 0 bits in −1 in order to reduce the unwanted output bias of the convolutional units when they receive a zero input.

### 4.3. The Proposed Architecture

The architecture of our classifier is a deep CNN with nine layers using Parametric Rectified Linear Units (PReLU) for feature extraction and a three-layer fully connected perceptron for actual classification. Figure 2a reports the layout of the proposed network.

The PReLU activation function adaptively learns the parameters of the rectifiers and improves accuracy at negligible extra computational costs. A learnable parameter α is introduced, and different neurons can have different parameters or a group of neurons can share one parameter.
(3)$$\begin{array}{cc} {\mathrm{PReLU}}_{i}\left(x\right) & =\left\{\begin{array}{cc} x & \;\mathrm{if}\;x>0\\ \alpha_{i}x & \;\mathrm{if}\;x\leq 0 \end{array}\right.\\ \; & =\max\left(0,x\right)+\alpha_{i}\min\left(0,x\right) \end{array}$$

If $\alpha_{i}=0$, then PReLU degenerates to ReLU; if $\alpha_{i}$ is a small fixed value (such as $\alpha_{i}=0.01$), then PReLU degenerates to Leaky ReLU (LReLU). In our work, $\alpha_{i}$ has been set constant at 0.25.

The network is trained on 7×1024×1 input tensors that represent seven 1024-long fingerprints stacked as the channels of a 1024×1 image. Multitarget bioactivity prediction is a *multiclass, multilabel* classification, which our classifier also has to assess if a ligand is active at the same time on different targets. As a consequence, the output is a *vector label* that is a binary vector where 1 s indicates bioactivity with respect to a particular target.

In line with the most recent CNNs, we implemented convolutional layers by using *Depthwise Separable Convolution* (DSC) [48] to reduce network parameters and lower the computational load. The classical convolution operator computes an element of the output tensor Y by applying a kernel K with spatial extent s×s and depth *d* to the input tensor X.
(4)$$Y_{i,j,k}=\sum_{l=1}^{s}\sum_{m=1}^{s}\sum_{n=1}^{d}X_{i-l,j-m,k-n}K_{l,m,n}$$

Here, we are using the proper index notation for convolution without kernel flipping. In DSC, *d*
*spatial* kernels $K_{\left(h\right)}^{S}$ with s×s size compute 1-depth convolutions, and a 1×1×d
*depth* kernel $K^{D}$ provides the final convolution output.
(5)$$\begin{array}{c} \begin{array}{cc} & Y_{i,j}^{\left(h\right)}=\sum_{l=1}^{s}\sum_{m=1}^{s}X_{i-l,j-m,h}K_{(h)\;l,m}^{S},\;h=1\ldots d\\ & Y_{i,j,k}=\sum_{n=1}^{d}Y_{i,j}^{\left(h-n\right)}K_{n}^{D} \end{array} \end{array}$$

It can be shown that DSC can reduce the number of parameters by a factor of $1/s^{2}$ for each layer: Our network was built using only 2,252,959 parameters, which is about a 1:25 ratio with the size of the CDK1-only classifier proposed in our previous work. Figure 2b reports the details of the implemented model.

Classification is achieved by using an MLP with 64/32/32 ReLU units per layer, respectively, while the output consists of 20 sigmoidal units because the probabilities of each class is independent from the other classes’ probabilities. For this reason, a *binary crossentropy* loss function has been used instead of the usual *categorical cross-entropy*. This choice is reasonable because the network performs a “multi-label”, “multi-class” classification task.

Training was conducted using a 10-fold cross-validation training scheme, where a classic strategy was adopted with an approximate 80%:10%:10% split for training, validation, and test set, respectively. A 1:100 active/inactive ratio compared to the less abundant class (646 active compounds) was maintained in three data sets.

Hyperparameter tuning was performed as a grid search for the following values. Depth separable convolution filters [1024, 512, 256, 128, 64, 32, 16] with zero padding were tested. The learning rates tested were in the range of [10${}^{-5}$–$10^{-1}$], where each tested value was 10-times the previous one. The batch sizes tested were in the range of [8–64]. Early sopping was used to identify the optimal number of training epochs, and model checkpoint was used to save the best model after each epoch. Hyperparameter optimization took 64 days and was performed by using an NVIDIA TITAN Xp GPU, 3840 CUDA Cores. Notwithstanding the complexity of the architecture, each training session took about 6 hours due to the efficiency of the DSC convolution operation.

## 5. Conclusions

We introduced EMBER, a novel molecular embedding aimed at improving the effectiveness of CNNs for VS tasks. The innovation in our approach consists in representing the ligand’s structure by several molecular fingerprints stacked as the channels of the input tensor. The key idea behind EMBER is that molecular fingerprints are computed by not only using the same algorithmic process, but also by using complementary information collected from the molecular structure such that they can be regarded as the “spectra” of a sort of molecular image.

We proved EMBER effectiveness by using a deep neural architecture for ligand multiclassification with respect to their bioactivity on twenty protein kinase targets. We achieved very satisfactory results with respect to the classification task; in general, we obtained a very high capacity model with a very small number of parameters.

Moreover, we presented an explainability analysis by feature attribution showing that only three molecular fingerprints play an active role in classification, which are FeatMorgan, Layered, and RDKIT. Our findings confirm very recent studies that outline the relevance of functional description Fingerprints (i.e., Pharmacophore-like) when addressing bioactivity classification, especially for kinase inhibitors.

## Figures and Tables

**Figure 1 ijms-23-02156-f001:**
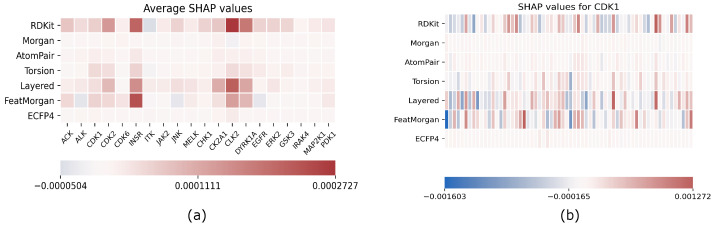
Explainability results using SHAP; (**a**) average SHAP values for each fingerprint computed on the entire test set separately for each target; (**b**) example of single target explainability analysis for CDK1: SHAP values are reported for each fingerprint, and each row has been grouped in 64 bins to enhance readability.

**Figure 2 ijms-23-02156-f002:**
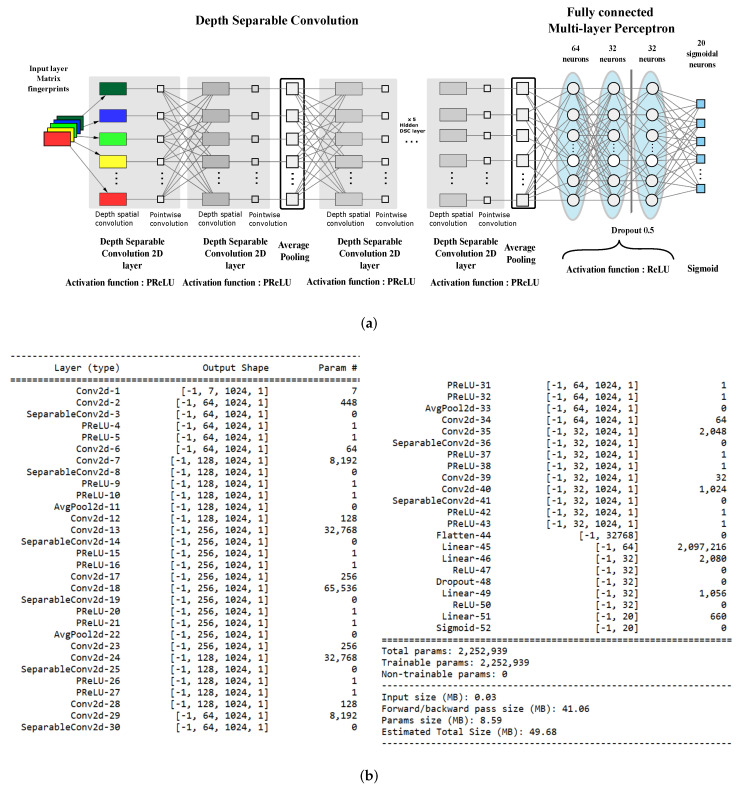
The proposed architecture. (**a**) Network layout. (**b**) Model summary.

**Table 1 ijms-23-02156-t001:** Accuracy metrics for all the targets. Best/worst values for each column are in bold/italic.

Target	Acc.	Loss	Sensitivity	MCC	AUC	F1-Score
ACK	0.9957	0.0226	0.5000	0.6742	0.9834	0.6463
ALK	0.9930	0.0402	0.6575	0.7913	0.9904	0.7804
CDK1	0.9910	0.0314	0.4537	0.6397	0.9850	0.6059
CDK2	0.9859	0.0431	0.5281	0.6338	0.9845	0.6287
CDK6	0.9966	0.0210	0.5865	0.7523	0.9895	0.7305
INSR	0.9893	0.0329	0.3779	0.5830	0.9858	0.5342
ITK	0.9945	0.0232	0.5886	0.7302	0.9905	0.7154
JAK2	0.9898	0.0472	**0.8474**	**0.9090**	**0.9950**	**0.9114**
JNK3	**0.9967**	**0.0154**	0.5905	0.7610	0.9901	0.7381
MELK	0.9957	0.0229	0.7081	0.8270	0.9897	0.8188
CHK1	0.9895	0.0512	0.6385	0.7650	0.9846	0.7565
CK2A1	0.9942	0.0253	0.5166	0.6944	0.9857	0.6667
CLK2	0.9936	0.0259	*0.2255*	*0.4137*	0.9771	*0.3485*
DYRK1A	0.9916	0.0321	0.4080	0.5987	0.9776	0.5591
EGFR	0.9845	*0.0604*	0.7536	0.8331	0.9874	0.8357
ERK2	0.9881	0.0563	0.7295	0.8292	0.9886	0.8272
GSK3	*0.9843*	0.0554	0.5827	0.6892	*0.9762*	0.6856
IRAK4	0.9936	0.0287	0.7611	0.8611	0.9938	0.8571
MAP2K1	0.9931	0.0319	0.5497	0.7184	0.9795	0.6954
PDK1	0.9945	0.0271	0.6310	0.7757	0.9875	0.7613

**Table 2 ijms-23-02156-t002:** True Positives versus Positives ratio and Enrichment Factors computed on the entire test set.

Protein	TP/P 1% *	TP/P 2% *	TP/P 5% *	TP/P 10% *	EF 1%	EF 2%	EF 5%	EF 10%
ACK	72/106	84/106	95/106	101/106	68	40	18	10
ALK	131/254	202/254	229/254	247/254	52	40	18	10
CDK1	111/205	150/205	189/205	196/205	54	37	18	10
CDK2	118/303	194/303	264/303	289/303	39	32	17	10
CDK6	79/104	90/104	98/104	101/104	76	43	19	10
INSR	110/217	145/217	195/217	206/217	51	33	18	9
ITK	107/158	125/158	148/158	155/158	68	40	19	10
JAK2	134/832	268/832	669/832	804/832	16	16	16	10
JNK3	81/105	88/105	95/105	102/105	77	42	18	10
MELK	130/185	157/185	178/185	181/185	70	42	19	10
CHK1	134/343	233/343	300/343	324/343	39	34	17	9
CK2A1	100/151	117/151	141/151	146/151	66	39	19	10
CLK2	59/102	73/102	87/102	96/102	58	36	17	9
DYRK1A	97/174	126/174	152/174	162/174	56	36	17	9
EGFR	134/702	268/702	586/702	664/702	19	19	17	9
ERK2	133/525	267/525	471/525	505/525	25	25	18	10
GSK3	132/393	226/393	327/393	353/393	34	29	17	9
IRAK4	134/339	263/339	320/339	333/339	40	39	19	10
MAP2K1	118/191	142/191	167/191	178/191	62	37	17	9
PDK1	123/187	149/187	170/187	181/187	66	40	18	10

* Percentage relative to the evaluated test set evaluated (13400 compounds), i.e., 1% = 134 molecules.

**Table 3 ijms-23-02156-t003:** The top five test set molecules prioritized by our classifier as the most active on the CDK1 target.

Molecule ChEMBLID	Chemical Structure	$IC_{50}$
CHEMBL192216	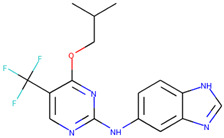	2 nM
CHEMBL3644025	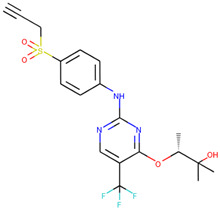	82 nM
CHEMBL445125	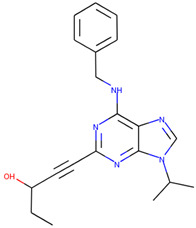	500 nM
CHEMBL2403087	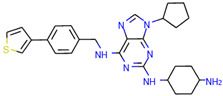	183 nM
CHEMBL2403084	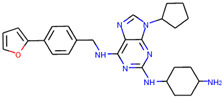	148 nM

**Table 4 ijms-23-02156-t004:** A summary of all proteins (active and inactive) obtained from preprocessing methods.

Target	PDB ID	Ligand Code *	Actives	Inactives
ACK	5ZXB	9KO	746	159,775
ALK	6E0R	HKJ	1665	227,247
CDK1	6GU2	F9Z	1241	124,473
CDK2	6INL	AJR	1924	225,087
CDK6	5L2S	6ZV	646	256,561
INSR	5E1S	5JA	1423	195,990
ITK	4RFM	3P6	1001	135,007
JAK2	6M9H	J9D	5526	577,409
JNK3	2B1P	AIZ	658	95,252
MELK	6GVX	TAK	1215	246,662
CHK1	6FC8	D4Q	2175	21,763
CK2a1	6JWA	5ID	1053	10,534
CLK2	6FYL	3NG	671	6800
DYRK1A	4YLK	4E2	1126	11,274
EGFR	5GNK	80U	4757	47,541
ERK2	6OPH	6QB	3525	35,237
GSK3B	5F94	3UO	2578	25,768
IRAK4	6EG9	OLI	2131	21,282
MAPK2K1	4AN9	ACP; 2P7	1254	12,508
PDK1	3NAX	MP7	1117	11,166

* Most affine lingands.

## Data Availability

Both the code developed in this research and the related data sets are available at the following GitHub repository: https://github.com/CHILab1/DSC-Multi-Fingerprint-Embedding (accessed on 9 January 2022).

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
