# Peer review of "EMBER—Embedding Multiple Molecular Fingerprints for Virtual Screening"

_ijms, 2022, doi:10.3390/ijms23042156_

Round 1
Reviewer 1 Report
In this paper, the authors developed a useful and novel molecular embedding model as a computational method for drug R&D. The application of the EMBER embedding might accelerate the identification of new ligands targeted kinases such as CDK1 in the future.
However, there is still some problems to be addressed before the publication in IJMS.
- Introduction: line 75, this study focused on the protein kinase CDK1, why authors mentioned the mutation on BRAF? Please give some explanation in this response.
- Table 1. The accuracy values of these twenty kinase were very close and JNK3 was the best, why authors pay attention to CDK1?
- Discussion: The authors had listed out some drawbacks of existed database in the Introduction, I recommended they critically described the highlighting and weakness of this study.
- Reference: Please notice the format of reference, for example, some journal names were full name while some were abbreviation; some references also missed some information.
- Option: Can authors give some predicted chemical ligands based on the database for kinase such as CDK1?
Author Response
Dear Reviewer,
First of all, we would like to thank you for the improvement suggestions to the manuscript.
The English was improved, and especially the text was appropriately edited where sentences were excessively long and/or unclear.
Below we report the responses point by point to simplify the reading.
- “Introduction: line 75, this study focused on the protein kinase CDK1, why authors mentioned the mutation on BRAF? Please give some explanation in this response.”
Authors Response: BRAF was used as an example of mutated kinase considered in oncology. We re-organised the sentences in order to better contextualise the choice of kinases as valuable target in cancer therapeutics (Line 65 to 84).
- “Table 1. The accuracy values of these twenty kinase were very close and JNK3 was the best, why authors pay attention to CDK1?”
Authors Response: As emphasized from line 78 to 84 the study focuses on the results obtained on CDK1 because this is the target our Drug Discovery Unit is working on.
- “Discussion: The authors had listed out some drawbacks of existed database in the Introduction, I recommended they critically described the highlighting and weakness of this study.”
Authors Response: From line 277 to 300 we reported a brief discussion on the creation of our dataset highlighting strengths and weaknesses.
- “Reference: Please notice the format of reference, for example, some journal names were full name while some were abbreviation; some references also missed some information.”
Authors Response: References have been corrected.
- “Option: Can authors give some predicted chemical ligands based on the database for kinase such as CDK1?”
Authors Response: We included an additional table (Table 3) showing the top five molecules prioritized on the test set by the model with respect to target CDK1. In the table we have included CHEMBLID, 2D molecular structure, and IC50. From line 215 to 219 we have included a brief commentary on the table.

Reviewer 2 Report
Dear authors,
Your work is adding a new approach to the VS arsenal with excellent statistical results and it would be interesting to monitor future applications of your work in drug-design efforts.
Best regards,
Author Response
Thank you very much for your comments.